# YOLOv8 Architectural Scene Section Recognition Method Based on SimAM-EMA Hybrid Attention Mechanism

**DOI:** 10.3390/s25103060

**Published:** 2025-05-12

**Authors:** Jiangang Ye, Zilong Shu, Wen Zhou, Weijun Hu, Jinwen Qiu, Feng Xu, Hui Wang, Guoliang Luo

**Affiliations:** 1Quzhou Special Equipment Inspection&Testing Research Institute, Quzhou 324000, China; 120749632@163.com (J.Y.); dester19831219@126.com (W.Z.); wayne1990red@hotmail.com (W.H.); 2School of Information and Software Engineering, East China Jiaotong University, Nanchang 330013, China; 2023218085405018@ecjtu.edu.cn (Z.S.); 2024068081200020@ecjtu.edu.cn (J.Q.); 2022029081100008@ecjtu.edu.cn (F.X.); luoguoliang@ecjtu.edu.cn (G.L.)

**Keywords:** YOLOv8, hybrid attention mechanism, architectural scene segmentation, SimAM, EMA

## Abstract

To overcome the limitations of current methods in architectural scene segmentation accuracy, this study presents a hybrid attention-enhanced YOLOv8 framework and introduces a dedicated building interior profile dataset. The proposed approach extends the YOLOv8 architecture by integrating SimAM (Simple, Parameter-Free Attention Module) to dynamically evaluate neuron significance and refine feature representations. This is coupled with an Efficient Multi-Scale Attention (EMA) module that synergizes local and global attention mechanisms, enabling robust multi-scale feature fusion while maintaining stable weight optimization. To address the scarcity of domain-specific data, a meticulously annotated dataset encompassing common architectural interior elements is developed. Comprehensive evaluations demonstrate that the enhanced model achieves an 89.9% mAP@0.5 on the proposed dataset, outperforming the baseline YOLOv8 with relative improvements of 4.5% in precision, 5.2% in recall, 5.1% in mAP@0.5, and 4.5% in mAP@0.5–0.95. These advancements underscore the efficacy of hybrid attention mechanisms in architectural scene analysis and establish a benchmark dataset to facilitate future research in intelligent building environment interpretation.

## 1. Introduction

With the rapid development of 3D acquisition, display, and plotting technology, 3D modeling has become one of the key technologies in computer-aided civil and infrastructure engineering. As a result, the application of large-scale 3D infrastructure models is becoming increasingly important in these industries.

While researchers have extensively investigated the basic techniques used for the feature extraction, segmentation, and rendering of conventional 3D models, there is an increasing demand in the industry for processing techniques dedicated to large-scale 3D infrastructure models. With the development of computer-aided civil engineering, the informatization of various requirements in civil engineering has become one of the mainstream trends.

In civil engineering, a sectional plan is mainly used to represent the internal structure of an object. It is performed by applying a cutting plane (plane or surface) to section the object, removing the part between the observer and the cutting plane, and projecting the rest onto the projection plane. The resulting diagram is referred to as a section view.

Especially in architecture, a sectional plan refers to the cross-section of a building, with one or more upright sections perpendicular to the axis of the exterior walls. The resulting projection is used to represent the internal structure or construction form of the building, along with the layering situation, the connections between each part, the materials and heights, etc.; the sectional plan is one of the most indispensable and essential drawings used in collaboration with the ichnography and elevation drawings. The 3D cutting (or 2D plane cutting) of sectional views is a critical processing technique for CID design analysis and the exploration of data distribution inside the structure, as introduced by Yang [1] et al. Especially for modern building informatization modeling and design, sectional drawings based on 3D cutting are essential references for observing the details and verifying the correctness of the design. For large-scale 3D infrastructure modeling, 3D cutaway-based profiles are necessary for designers’ evaluation and valuable for engineers to review the 3D details. For example, sectional plans can provide an opportunity to perceive the correctness of 3D models enclosed in larger 3D objects.

With the advancement of 3D modeling technology, building information modeling (BIM) [2] has been widely used to produce various 3D building models. Through the 2D plane cutting of 3D building models, the section view of buildings can be efficiently and accurately obtained. Further, by introducing advanced image recognition algorithms, we can automatically analyze and identify each component and detail in the profile. This method improves the accuracy and efficiency of identification. It provides new ideas for extending the identification to the section plane of a building floor and even to the component identification in the section plane of large substation models. Architectural section detection differs significantly from general object detection tasks. Firstly, architectural section images have special structural characteristics; they contain rich geometric shapes, material boundaries, and spatial relationship information, along with multi-scale features, requiring precise identification of both large structural components (such as walls, floor slabs) and smaller detail components (such as doors, windows, furniture). Secondly, architectural section images typically have high information density, with complex layering and nesting relationships between various components, increasing the difficulty of detection.

In the field of computer vision, attention mechanisms have become a key technology for improving model performance by mimicking the selective attention process of the human visual system, enabling models to focus on the most relevant parts of input data. In architectural scene analysis, attention mechanisms are particularly important due to the complex geometric shapes and multi-level spatial relationships in structures. Addressing the unique challenges of architectural section detection, SimAM precisely captures critical lines and geometric features by evaluating neuronal importance, while EMA enhances the understanding of spatial relationships between components by integrating local and global information. The combination of these mechanisms not only improves the model’s ability to recognize complex architectural elements but also enhances the efficiency of multi-scale feature processing, enabling more accurate identification of various components in architectural sections, from small details to large structural elements, effectively addressing detection challenges across different scales.

The main contributions of this paper are as follows: (1) A dataset of architectural section planes was created, which provides ideas for the subsequent identification of components of 3D model cutting planes, such as large-scale building scenes and substations. (2) A hybrid attention mechanism based on quantitative evaluation was proposed. The improved method is more accurate than the original YOLO series model and MM detection model.

## 2. Related Work

The use of 3D cutting in section-based analysis has been widely studied in various fields, such as manufacturing and medical analysis based on 3D models. Specifically, sectional views are an important tool for design optimization and the visualization of large structural models [3].

A profile is mainly used to represent an object’s internal structure. It is obtained by applying a sectional plane (plane or surface) to cut the object, removing the part between the observer and the sectional plane, and projecting the rest onto the projection plane. The resulting figure is called the section view. Section views have always been an important reference point for designing and analyzing project structures and construction drawings. Holgado-Barco [4] et al. proposed a method for processing the data captured by a moving laser scanning system to obtain the geometric cross-section of a highway. Arias [5] et al. used photogrammetric information to collect the sectional properties of an ancient, irregular piece of wood, further analyzed the geometric data on this basis, and finally realized the three-dimensional reconstruction of the scanned object. Based on the important information in the profile, Lin [6] et al. developed a safe structural form and construction method for underground infrastructure. Karhu [7] et al. developed a general construction process modeling method based on a local view of the model. This approach is particularly valuable for industry practitioners, enabling them to interact with the model and represent scheduling in the construction process. Yuan [8] et al. determined the location of the section plane through medical anatomical points, using a rotation sequence of the long-axis and short-axis views of the anchor points to conduct a three-dimensional reconstruction of the heart. Hu H. [9] et al. analyzed organ profiles to classify organs based on color blocks and reconstructed three-dimensional organs by multi-angle projection, filter-back projection, and other strategies. Shi [10] et al. proposed a deep learning framework, MsvNet, based on a multi-sectional view (MSV) representation for feature segmentation and recognition algorithms. Y. Wen [11] et al. used a profile to reconstruct a 3D model from a 2D profile by enhancing the traditional volume-based method. By locating the object recognition geometric plane of the profile, Gong [12] et al. developed a graphical representation to describe multiple relationships between individual views in a two-dimensional graphical space and implemented an evidence theory-based reasoning technique to validate the relationships used to fold views and profiles in a three-dimensional object space.

In recent years, significant progress has been made in target recognition through the adoption of deep learning techniques, transitioning from traditional feature-based manual algorithms to automatic recognition methods based on deep neural networks [13]. Currently, deep learning technology has made remarkable achievements in the field of target detection. In particular, convolutional neural networks (CNNs), proposed by LeCun et al., first demonstrated their superior performance in handwritten digit recognition tasks [14]. Among them, the YOLO (You Only Look Once) family of models is a class of deep learning models widely used in target detection, with the unique feature of end-to-end detection, enabling the simultaneous completion of target detection and localization in a single forward propagation, with the advantages of high efficiency and accuracy. There have been multiple studies on the application of YOLO series models in the construction industry. Kisaezehra [15] et al. implemented a safety helmet detection system at construction sites, which utilized the YOLOv5 model to achieve real-time detection of safety helmets at construction sites, providing technical support for ensuring worker safety. Feng [16] et al. explored the application of YOLO models in identifying safety risks at construction sites. Wang [17] et al. published ’YOLOv7 Optimization Model Based on Attention Mechanism Applied in Dense Scenes,’ which optimized the YOLOv7 network by combining standard convolution with attention mechanisms, proposing a YOLOv7B-CBAM model that effectively improved object detection accuracy in dense construction scenarios. These studies indicate that the YOLO series models, due to their real-time capability and high accuracy, have broad application prospects in the construction industry, while the introduction of attention mechanisms can further enhance their recognition capabilities in complex construction scenes.

Therefore, it can be inferred that the sectional plan obtained by a 3D model cutting method has significant research value in various fields.

## 3. Methodology

With the development of information technology in construction, the intelligent generation of building sectional plans and the recognition of sectional information are significant. The recognition of a building model’s sectional plan allows engineers to examine 3D details and evaluate the safety and rationality of the building [18]. A hybrid attention mechanism based on the YOLOv8 model is proposed for the current 3D architecture. This attention mechanism dramatically improves the recognition accuracy of the dataset of the internal profile plane of the building. Firstly, the input images are normalized, and then the feature extraction and other operations are carried out through the backbone network. Finally, the image is detected through the detection head network. The overall workflow of the network is as follows, see Figure 1:

### 3.1. YOLOv8 Model

The YOLO series model was proposed by Joseph Redmon et al. The original YOLOv1 used a single convolutional neural network to directly predict bounding boxes and categories, achieving efficient object detection. With each iteration, the YOLO model continues to introduce new technologies and improvements, such as the batch normalization of YOLOv2 [19], the multi-scale prediction of YOLOv3 [20], CSPDarknet53 [21], the architecture of YOLOv4 [22], and the lightweight design of YOLOv5. Building on this, YOLOv8 is further optimized by enhancing the model architecture and adopting more efficient training strategies and optimization methods, resulting in significantly improved detection accuracy and speed. The architecture design of YOLOv8 is mainly reflected in the following aspects:

#### 3.1.1. Improved Feature Extraction Network

YOLOv8 has significantly improved feature extraction networks, adopting a deeper and wider network structure—CSPNet (Cross Stage Partial Network)—to improve the processing capability of complex scenes. Implementing CSPNet effectively reduces the computational cost and improves the model’s feature representation ability. CSPNet [23] reduces redundant computation by passing partial features layer by layer and fusing these features at specific layers. YOLOv8 uses improved Backbone networks like CSPDarknet53 [24] to enhance its feature extraction capabilities. The new Backbone network improves the depth and width of the model by adding convolutional layers and optimizing the residual structure.

#### 3.1.2. Multi-Scale Feature Fusion

YOLOv8 introduces multi-scale feature fusion techniques, such as FPN (Feature Pyramid Network) [25], which improves the detection accuracy of small and large targets by constructing bottom-up feature pyramids and combining feature maps at different scales, and PANet (Path Aggregation Network) [26], which further enhances the richness of feature expression and the detection accuracy of targets at different scales by top-down path augmentation feature fusion [27].

#### 3.1.3. New Activation Function

YOLOv8 uses the Mish [28] activation function, which performs better in training deep neural networks than traditional ReLU functions. Compared with ReLU, the Mish function has better smoothness and nonlinear characteristics, which helps to improve the model’s expression ability and training stability.

#### 3.1.4. Attention Mechanism

YOLOv8 introduces the SE (Squeeze and Excitation) [29] module to improve detection accuracy by focusing on important features. The SE module adjusts the weight of the feature map by global information so that the model can pay more attention to important features and improve the detection performance.

### 3.2. SimAM Attention Mechanism

The SimAM [30] attention mechanism is a simple but very effective attention module for convolutional neural networks. In contrast with the existing channel and spatial attention modules, SimAM does not require additional parameters to the original network. Instead, it infers the 3D attention weights of the feature graph in a single layer. Specifically, SimAM, based on some well-known neuroscience theories, proposes to optimize an energy function to discover the importance of each neuron, further deduces a solution for a fast closed form of the energy function, and shows that the solution can be implemented in less than ten lines of code. Another advantage of this module is that most operators are selected according to the solution of the defined energy function, avoiding too much effort on the structural adjustment. Through the quantitative evaluation of various visual tasks, the flexibility and effectiveness of the module have been demonstrated, and the expressibility of many ConvNets has been improved.

The method defines the following energy function:(1)$$\begin{array}{c} e_{t}\left(w_{t},b_{t},y,x_{i}\right)={\left(y_{t}-\left(\hat{t}\right)\right)}^{2}+\frac{1}{M-1}\sum_{i=1}^{M-1}{\left(y_{0}-\hat{x_{i}}\right)}^{2} \end{array}$$
where $\hat{t}=w_{t}t+b_{t},\hat{x_{i}}=w_{i}x_{i}+b_{t}$ denotes the linear transformation of *t*; $w_{i}$ are the target neurons and the other neurons in a single channel of the input feature $X\in R^{C\times H\times W}$; i is the index in the spatial dimensions; M=H×W is the number of neurons in this channel; and $w_{t}$ and $b_{t}$ are the weights and biases for the transformation. When n=yt, Formula (1) reaches its minimum value, and all other $n=y_{0}$, where $y_{t}$ and $y_{0}$ are two different values. By minimizing the equation, Formula (1) is equivalent to finding the linear separability between the target neuron t and all other neurons within the same channel. For simplicity, we adopt binary labels (i.e., 1 and −1) for $y_{t}$ and $y_{0}$ and add a regularizer to Equation (Equation 1). The final energy function is as follows:(2)$$\begin{array}{c} e_{t}\left(w_{t},b_{t},y,x_{i}\right)=\frac{1}{M-1}\sum_{i=1}^{M-1}{\left(-1-\left(w_{t}x_{i}+b_{t}\right)\right)}^{2}+\;{\left(1-\left(w_{t}t+b_{t}\right)\right)}^{2}+\lambda w_{t}^{2} \end{array}$$

Theoretically, each channel has M energy functions. Solving all these equations with iterative solvers like SGD is a computational burden. Fortunately, Formula (2) has a fast closed solution, which can be easily resolved using the following:(3)$$\begin{array}{c} w_{t}=-\frac{2(t-\mu_{t})}{{\left(t-\mu_{t}\right)}^{2}+2\sigma_{t}^{2}+2\lambda } \end{array}$$(4)$$\begin{array}{c} b_{t}=-\frac{1}{2}\left(t+\mu_{t}\right)w_{t} \end{array}$$

Since all the neurons on each channel follow the same distribution, the mean and variance can be computed first for the input features in both the H and W dimensions, while avoiding repeated computations, as follows:(5)$$\begin{array}{c} e_{t}^{*}=\frac{4({\hat{\sigma }}^{2}+\lambda )}{{\left(t-\hat{\mu }\right)}^{2}+2{\hat{\sigma }}^{2}+2\lambda )} \end{array}$$
where $\hat{\mu }=\frac{1}{M}\sum_{i=1}^{M}x_{i},{\hat{\sigma }}^{2}=\frac{1}{M}\sum_{i=1}^{M}{\left(x_{i}-\hat{\mu }\right)}^{2}$. The lower the energy $e_{t}^{*}$ of a neuron t, the greater difference between it and the surrounding neurons, and the more important it is for visual processing. Therefore, the importance of each neuron can be determined by $\frac{1}{e_{t}^{*}}$

The whole process can be expressed as follows:(6)$$\begin{array}{c} \tilde{X}=sigmoid\frac{1}{E}⊙X \end{array}$$
where *E* groups all $\frac{1}{e_{t}^{*}}$ along the channel and spatial dimensions. Adding a sigmoid function limits the values in *e* that are too large. The sigmoid is a monotonic function, so it does not affect the relative importance of each neuron.

The SimAM attention mechanism can infer three-dimensional attention weights for feature graphs without increasing the original network parameters. It is proposed to optimize the energy function to tap into each neuron’s importance.

### 3.3. EMA Attention Mechanism

The EMA [31] attention mechanism is a novel and efficient multi-scale attention mechanism that focuses on preserving information about each channel and reducing computational overhead by realigning some channels into batch dimensions, as well as grouping channel dimensions into multiple sub-features, so that the spatial semantic features are evenly distributed within each feature group. Specifically, in addition to encoding the global information in each parallel branch to recalibrate the channel weights, the output features of these two parallel branches are further aggregated through cross-dimensional interactions to capture pixel-level pair relationships, addressing the potential side effects of modeling cross-channel relationships through channel dimension reduction on extracting depth vision representations.

The EMA attention mechanism optimizes the representation of multi-scale features by introducing the methods for examining cross-dimensional interaction and feature grouping without significantly increasing the computational complexity. It also provides an efficient solution for inferring three-dimensional attention weights from feature graphs. The mechanism enables the following processes:

1. Feature grouping: The input feature map is divided into multiple sub-features according to channel dimensions to learn different pieces of semantic information.

2. Parallel subnetworks: The attention-weight descriptors of feature graphs are extracted by parallel processing of 1 × 1 and 3 × 3 convolution. The 1 × 1 branch encodes channel information through two 1D global averaging pooling operations, while the 3 × 3 branch captures multi-scale features through a 3 × 3 convolution.

3. Cross-space learning: A 2D global average pooling operation encodes global spatial information, and a Softmax nonlinear function fits a Gaussian distribution to generate a spatial attention map.

The global averaging pooling formula for input features with width W is as follows:(7)$$\begin{array}{c} z_{C}^{W}\left(W\right)=\frac{1}{H}\sum_{0\leq j\leq H}x_{c}\left(j,W\right) \end{array}$$
where *C* denotes the number of input channels; *H* and *W* represent the spatial dimensions of the input features, respectively; and $x_{c}$ denotes the input feature of the c-th channel.

The formula for the 2D global averaging pooling operation is as follows:(8)$$\begin{array}{c} z_{C}=\frac{1}{H\times W}\sum_{i=0}^{H}\sum_{j=0}^{W}x_{c}\left(i,j\right) \end{array}$$

It is designed for encoding global information and modeling remote dependencies. A natural nonlinear function, Softmax with 2D Gaussian mapping, is adapted to fit the aforementioned linear transformation at the output of 2D global average pooling to enhance computational efficiency. We obtain the spatial attention map by multiplying the production of the above-mentioned parallel processing with the matrix dot product operation.

The EMA attention mechanism captures pixel-level pairing relationships and highlights the global context by collecting multi-scale spatial information at the same processing stage. The final output feature map is the same size as the input and can be efficiently stacked into modern architectures. At the same time, the EMA attention mechanism effectively fuses cross-channel and spatial position information to enhance the pixel-level attention of convolutional neural networks on high-level feature maps, providing significant performance improvements for visual tasks.

### 3.4. Hybrid Attention Mechanism

As mentioned, the SimAM [32] attention mechanism is a simple but very effective attention module for convolutional neural networks. Based on neuroscience theory, it optimizes the energy function to discover the importance of each neuron without adding parameters to the original network. In addition, the EMA attention mechanism is an efficient multi-scale attention module that enhances spatial semantic feature distribution by reshaping channels into batch dimensions and grouping them into sub-features. It also utilizes interdimensional interactions to capture pixel-level relationships. See Figure 2.

To perform feature extraction, input feature maps are processed by the SimAM and EMA modules. The SimAM module optimizes the energy function, discovering the importance of each neuron, while the EMA module optimizes multi-scale feature representation through reshaping and grouping operations. Using the fused feature map can improve the feature representation capability without significantly increasing the computational complexity. This hybrid mechanism can take advantage of the simple efficiency of SimAM and the multi-scale feature representation of EMA to provide a richer feature representation. The fusion process is shown in Figure 3.

Firstly, the SimAM module receives the image features and deduces the 3D attention weights for the feature graphs. Then, the EMA module receives image features reweighted by SimAM, performs grouping and parallel convolution processing on them, and outputs the final image features. This hybrid attention mechanism takes full advantage of the parametric efficiency and simplicity of SimAM, as well as EMA’s multi-scale feature representation capabilities and computational efficiency.

We can express the combination of the SimAM and EMA modules as follows:(9)$$\begin{array}{c} Hybrid(X)=\alpha \cdot SimAM(X)+\beta \cdot EMA(X) \end{array}$$
where α and β are the fusion weights, adjusting the influence of SimAM and EMA in the overall feature map. According to the above-mentioned calculation formula, the hybrid attention mechanism can improve the feature representation ability without significantly increasing the computational complexity and is thus suitable for various visual tasks.

## 4. Experimental Analysis

This study employs a YOLOv8 architecture based on hybrid attention mechanisms for architectural scene section recognition. The experimental procedure primarily consists of the following four stages: dataset creation, model architecture design, model training and optimization, and performance evaluation.

### 4.1. Building Scene Internal Profile Dataset Production

Since there is limited research on identifying profiles within architectural scenes, standardized datasets for testing scene profiles are scarce. Therefore, we built some models of the interior of the building using Blender 4.1.2 and other software, incorporating various furniture commonly found in the interior as an example. We generated accurate building interior section images by setting up cutting planes perpendicular to the axis in the three-dimensional model, removing the portion between the observer and the cutting plane, and projecting the remaining portion onto the projection plane. This method provides a clear representation of the building’s internal structure and a consistent data foundation for model training.

When creating the dataset, we followed the following principles: 1. The location is reasonable and in line with human cognition. For example, the models can be physically adjacent, but no collision should result in the table inside the bed, which is not in line with reality. 2. In line with the actual needs of the field of civil engineering construction, only the X, Y, and Z axes are cut because, in practical applications, the model only needs to be cut along the X, Y, and Z directions to meet the needs of civil engineering analysis of the section plane; 3. Since the purpose of section plane identification in this work is to further extend to the identification of buildings, industrial machines, and other equipment; these tools or buildings usually do not have pronounced color characteristics. Unified decolorization is performed on all furniture models to ensure that color-related issues do not affect identification accuracy.

### 4.2. Implementation Details

The experimental setup in this paper uses Windows 10 as the operating system, an NVIDIA RTX 3060 GPU device, Blender 3.6 for building the scene cross-sectional dataset, and PyTorch 2.1.0 for constructing the experimental code. The initial learning rate for training was set to 0.01, with a weight decay of 0.005, and the number of epochs for training on the scene cross-section dataset was 300. The input image size was 640 × 640 pixels.

### 4.3. Comparative Experiment of Attention Mechanism

To verify the effectiveness of the hybrid attention mechanism, we introduced the performance of several different attention mechanisms, such as SimAM, EMA, GMA [33], SEA [34], and CA [35], and verified each of them in turn; all conditions were held constant. YOLOv8 was used to compare the different attention mechanisms. The experimental results are shown in Table 1, Table 2 and Table 3.

According to the experimental results in Table 1, adding the EMA attention mechanism improves the accuracy and mAP@0.5. In contrast, adding the SimMA attention mechanism improves the recall rate. To effectively identify the influence of these attention mechanisms on the detection results, the two attention mechanisms were randomly mixed, and the optimal combination was obtained as SimAMEMA, which was superior to the baseline index across all indicators.

The results show that our hybrid attention achieves a 4.5% improvement over the baseline accuracy, a 5.2% improvement over recall, a 5.1% improvement over MAP@0.5, and a 4.5% improvement over the average accuracy of mAP@0.5:0.95. Therefore, the model’s detection performance has improved. In addition, we compare our improved network of mixed attention mechanisms with several other advanced networks, as shown in Table 3:

### 4.4. Ablation Experiment

To evaluate the effectiveness of the mixed attention mechanism, we conducted ablation experiments on the baseline model using the scene profile dataset that we produced.

Based on the results shown in Table 4, the ablation experiments highlight the importance and effectiveness of integrating each module. Notably, the combination of the SimAM and EMAv2 modules improved the overall accuracy of YOLOv8.

## 5. Conclusions

In this paper, we proposed a hybrid attention mechanism based on the YOLOv8 architectural scene profile recognition method, which can optimize energy function to discover the importance of each neuron, enhance spatial semantic feature distribution by reshaping channels into batch dimensions and grouping them into sub-features, and capture pixel-level relationships using interdimensional interactions. The hybrid attention-enhanced YOLOv8 model proposed in this study achieved significant results in architectural section recognition, which has broad application prospects. In the field of architectural design, the model can be used in BIM (Building Information Modeling) systems to automatically identify component elements in architectural section drawings, improving design efficiency and accuracy. In terms of building safety monitoring, the model can be used to detect potential safety hazards in building structures in real time, such as structural deformation or non-compliant designs. The experiments show that, compared with the existing methods, our proposed mixed attention mechanism performs better in realizing the architectural scene profile dataset, achieving an mAP@0.5 of 89.9%. Precision, recall, mAP@0.5, and mAP0.5–0.95, respectively, increased by 4.5%, 5.2%, 5.1%, and 4.5%, compared with the baseline of YOLOv8 model.

Although this study has achieved good results in architectural section recognition, there are still some limitations in practical applications. First, the current model is mainly optimized for conventional building interior objects, and its recognition accuracy may decrease when applied to unconventional building structures or special components. Second, the model’s performance may be compromised in highly complex sectional images or under extremely poor lighting conditions. See Figure 4.

## Figures and Tables

**Figure 1 sensors-25-03060-f001:**
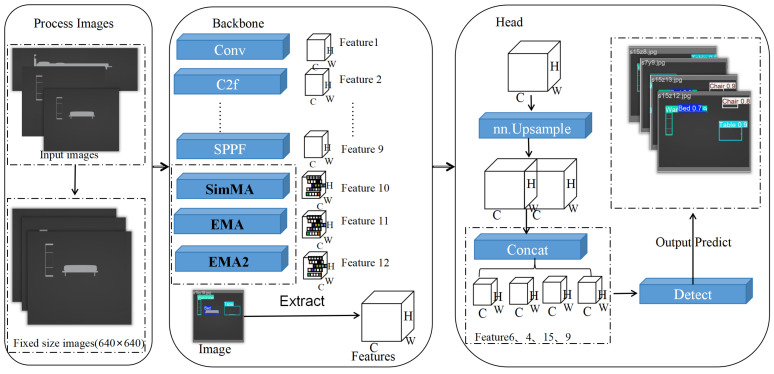
Workflow of YOLOv8 architectural scene profile recognition method based on hybrid attention mechanism.

**Figure 2 sensors-25-03060-f002:**
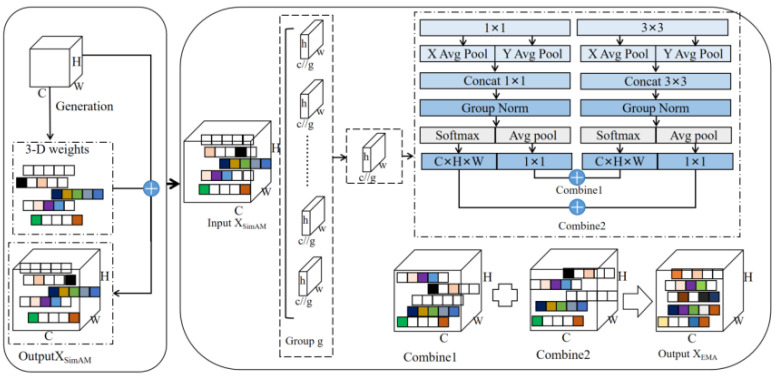
Hybrid attention mechanism module workflow.

**Figure 3 sensors-25-03060-f003:**
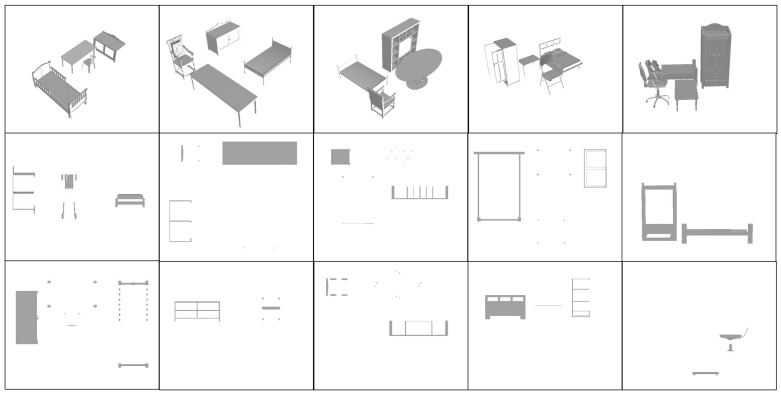
Architecture scene section plan dataset production.

**Figure 4 sensors-25-03060-f004:**
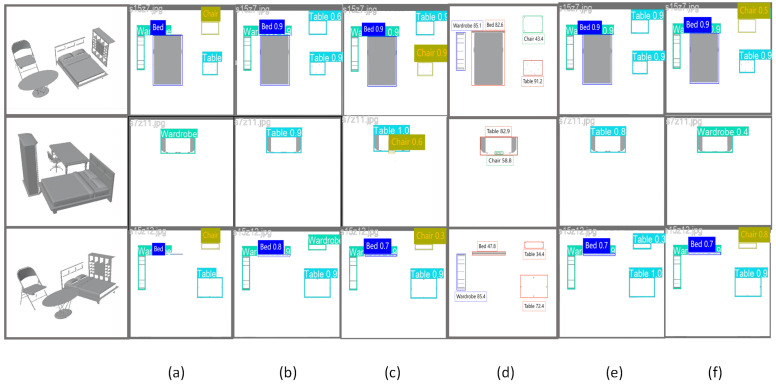
Comparison of five models. (**a**) Original image; (**b**) YOLOv5; (**c**) YOLOv8; (**d**) MMDetection; (**e**) YOLOv6; (**f**) Our model.

**Table 1 sensors-25-03060-t001:** Compares the detection ability of different attention mechanisms based on YOLOv8.

Attention	Precision	Recall	mAP@0.5	mAP@0.5:0.95
YOLOv8.yaml	78.2%	79.3%	84.8%	72.8%
SimAM	77.1%	82.3%	83.6%	71.5%
EMA	86.8%	78.6%	88%	74.8%
GAM	80.6%	77.4%	84.2%	72.6%
SEA	84.8%	77.4%	85.7%	73.9%
CA	83.7%	77.8%	84.4%	72.7%

**Table 2 sensors-25-03060-t002:** Comparison of detection capabilities of hybrid attention mechanisms.

Attention	Precision	Recall	mAP@0.5	mAP@0.5:0.95
YOLOv8.yaml	78.2%	79.3%	84.8%	72.8%
SimAM + EMA (Ours)	82.7%	84.5%	89.9%	77.3%

**Table 3 sensors-25-03060-t003:** Comparison of different model detection capabilities.

Model	Precision	Recall	mAP@0.5	mAP@0.5:0.95
YOLOv5	82.4%	72%	81%	68.3%
YOLOv8	78.2%	79.3%	84.8%	72.8%
MMDetection	74.4%	79.7%	87.7%	74.4%
YOLOv6	82.9%	78%	86%	73.4%
SimAM + EMA (Ours)	82.7%	84.5%	89.9%	77.3%

**Table 4 sensors-25-03060-t004:** Ablation experiments of mixed attention mechanism.

SimAM	EMAv1	EMAv2	Precision	Recall	mAP@0.5	mAP@0.5:0.95
—	—	—	78.2%	79.3%	84.8%	72.8%
✓	—	—	77.1%	82.3%	83.6%	71.5%
—	✓	—	86.8%	78.6%	88%	74.8%
—	—	✓	73.6%	86.3%	83.1%	72%
✓	✓	—	83.1%	78.5%	83.4%	71.8%
✓	—	✓	82.7%	84.5%	**89.9%**	**77.3%**

## Data Availability

The data presented in this study are available on request from the corresponding author.

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
