# Peer review of "YOLOv8 Architectural Scene Section Recognition Method Based on SimAM-EMA Hybrid Attention Mechanism"

_sensors, 2025, doi:10.3390/s25103060_

Round 1
Reviewer 1 Report
Comments and Suggestions for Authors
This article proposes a hybrid attention-enhanced YOLOv8 framework that dynamically evaluates the importance of neurons and optimizes feature representations by integrating SimAM (Simple, Parameter-Free Attention Module) with the aim of achieving robust multiscale feature fusion while maintaining stable weight optimization. Although, this research is highly valuable in the applied research of object detection and image segmentation modeling, however, this article is not without its drawbacks and to help this research in depth, I have some suggestions below.
Title, Abstract and Literature Section
1,. The title lacks a description of the highlights of the study and lacks precision, so it is recommended that it be modified to express the highlights of the study.
2,. The introduction of the background lacks the importance of the mechanism of attention.
3,. The literature section lacks research related to the re-building industry with YOLO.
Methods and materials section
1. It is recommended to focus on describing the steps of the experiment
2. How the internal section of the building was obtained or created needs to be further described
Conclusion and discussion section
- Lack of contribution to the description of the applications resulting from the results.
2, The constraints on the application also need to be explained specifically.
Author Response
Dear Reviewer,
Thank you for your valuable comments. Please refer to the attached response document for detailed replies.

Reviewer 2 Report
Comments and Suggestions for Authors
The article is of some interest as an attempt to use attention mechanisms in combination with YOLO-type networks. A comparative analysis of several variants of such mechanisms is given. This analysis is carried out for the task of segmentation of architectural scenes, but the given material can be adapted for the needs of solving other types of problems.
At the same time, reading the article causes a number of remarks, which are given below.
- The headings of subsections 3.2 and 3.3 coincide (SimAM Attention Mechanism), although 3.3 refers to the EMA attention mechanism. This appears to be a misprint.
- A large part of the elements in Figures 3 and 4 are practically indistinguishable and, consequently, it is very difficult to understand what is represented on them. It is necessary to give a version of them more suitable for visual perception.
- The reported experimental results look not convincing. It is necessary to give more examples with appropriate graphic presentation, as well as to give a more detailed analysis of the results obtained.
Overall conclusion: the article is of interest and can be published in Sensors after the above-mentioned shortcomings are eliminated.
Author Response

(The authors gave the same response as above.)

Reviewer 3 Report
Comments and Suggestions for Authors
As the main contribution of this work, two attention modules are utilized in the YOLOv8 network for architectural scene section detection, which is simple "A+B". Thus the novelty of the work is not sufficient.
The study background is not clear. What are the characteristics of architectural scene detection? What is different from other detection tasks?
How does the introduction of the attention modules help such a detection task?
Besides, some nouns in the manuscript are puzzling or nonstandard, e.g., MMdetection, etc.
Author Response

(The authors gave the same response as above.)

Round 2
Reviewer 1 Report
Comments and Suggestions for Authors
The authors have responded to my questions, and in addition, I suggest describing the parameters of the experimental equipment and the associated software.
Author Response

(The authors gave the same response as above.)

Reviewer 3 Report
Comments and Suggestions for Authors
All the concerned issues have been solved.
Author Response

(The authors gave the same response as above.)
